# Not All Noises Are Created Equally:
# Diffusion Noise Selection and Optimization

## Abstract

Diffusion models that can generate high-quality data from randomly sampled Gaussian noises have become the mainstream generative method in academia and industry. Are randomly sampled Gaussian noises equally effective for diffusion models? Some methods explore the impact of noise variations on the results, but they either do not operate in the pure noise space, requiring additional evaluation models, or cannot be adapted to general text-to-image tasks. In this paper, we mainly made three contributions. First, we are the first to hypothesize and empirically observe that the generation quality of diffusion models significantly depends on the noise inversion stability. This naturally provides a noise quality metric for noise selection, grounded in a mathematical property. Second, we further propose a novel noise optimization method that actively enhances the inversion stability of arbitrary given noises. Our method is the first one that purely optimizes noises for the general text-to-image task without relying on any additional evaluator or specifically designed prompts. Third, our extensive experiments demonstrate that the proposed noise selection and noise optimization methods both significantly improve representative diffusion models, such as SDXL and SDXL-turbo, in terms of human preference and other objective evaluation metrics. For example, the human preference winning rates of noise selection and noise optimization over the baselines can be up to **57%** and **72.5%**, respectively, on DrawBench.

## 1 Introduction

Generative diffusion models, renowned for the impressive performance (Dhariwal & Nichol, 2021), serve as the mainstream generative paradigm with wide applications in image generation (Nichol et al., 2021; Zhang et al., 2023; Saharia et al., 2022), image editing (Qi et al., 2023; Kawar et al., 2023), 3D generation (Gupta et al., 2023; Erkoç et al., 2023), and video generation (Ho et al., 2022a;c). Diffusion-based Generative AI products attracted much attention and a large number of users in recent years. Understanding and improving the capabilities of diffusion models has become an essentially important topic in machine learning.

It is well known that diffusion models can generate diverse results, which, of course, contain good ones and bad ones. Previous studies mainly enhance the generated results by working on model weight and architecture space (Song et al., 2020; Fang et al., 2023; Podell et al., 2023; Sauer et al., 2023; Ho et al., 2022b; Lin et al., 2024), while the noise space is largely overlooked. In this paper, we focus on the noise space. Some methods (Karthik et al., 2023; Ben-Hamu et al., 2024; Wallace et al., 2023) tried to select the results (equivalent to selecting initial noises) or optimize noise values with extra information, such as an additional image quality evaluator (Kirstain et al., 2023; Xu et al., 2024) or token IDs, which are used to construct a "noise-prompt" attention loss (Guo et al., 2024; Chefer et al., 2023; Agarwal et al., 2023) to improve the results in terms of visual effects and alignment. First, relying on an external evaluator introduces bias and limits generalization beyond the evaluator data. Since the evaluator's scores don't directly affect the noise, gradient backpropagation through the network is needed for optimization, which increases memory usage. Second, although the "noise-prompt" attention loss directly influences the noise, it only applies to specifically designed prompts, such as the $A+B$-type prompts, rather than handling real-world general prompts, including *style*, *detail description*, and *counting*, as shown in Figure 1.

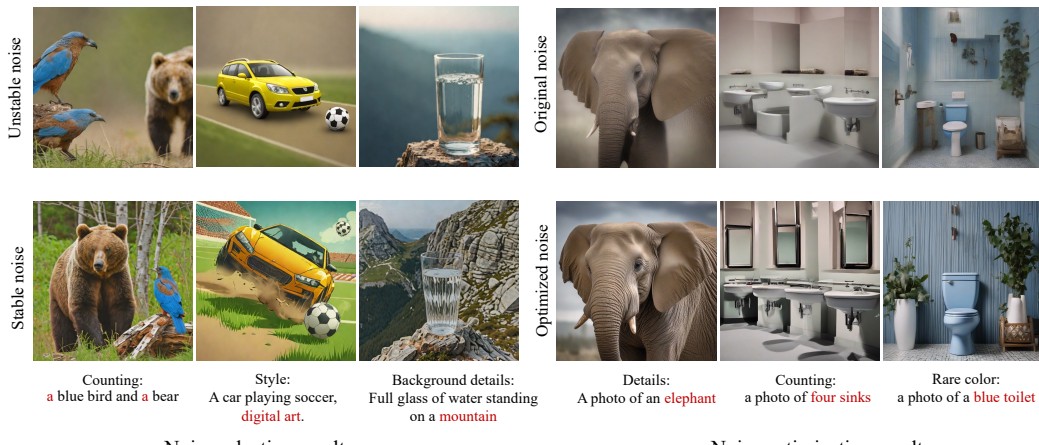

Figure 1: The qualitative results of noise selection and noise optimization. Left: SDXL-turbo. Right: SDXL. The proposed methods make improvements in multiple aspects.

In this work, we address two fundamental issues in the noise space of diffusion models. First, can a better noise be selected based on a mathematical metric rather than relying on external evaluation results? Second, is it possible to directly optimize a given noise to produce improved results using a general prompt and without any additional information? The answer to both questions is affirmative. Fortunately, we not only confirm the possibility but also propose practical algorithms.

**Contributions.** We summarize the three main contributions of this work as follows:

**First**, we hypothesize and empirically verify that not all noises are created equally. Specifically, random noises with high inversion stability usually lead to better generation than noises with lower inversion stability. The inversion stability measures the similarity of the sampled initial noise $\epsilon$ and the inverse noise $\epsilon'$. The mathematical quantitative metric naturally provides us with a novel noise selection method to select stable noises, which often correspond to better results. Unlike other methods that judge noise quality by image quality (we call this post-selection), we directly select the noise without introducing an additional evaluator. We present several qualitative results of noise selection in Figures 1 and 5.

**Second**, we further proposed a novel noise optimization method that actively enhances the inversion stability of arbitrary given noises. More specifically, we optimize an inversion-stability loss via gradient descent with respect to the sampled noise (rather than the conventional model weight space). The proposed noise optimization method is the first one that purely optimizes noises for the general text-to-image task, reducing memory usage while achieving better visual effects and alignments with general prompts. We present several qualitative results of noise optimization in Figures 1 and 7.

**Third**, our extensive experiments demonstrate that the proposed noise selection and noise optimization methods both significantly improve representative diffusion models, such as SDXL and SDXL-turbo. On the one hand, the human preference winning rates of noise selection and noise optimization over the baseline can be up to 57% and 72.5%, respectively, on DrawBench in terms of human preference. On the other hand, noise selection and noise optimization are also preferred by Human Preference Score (HPS) v2 (Wu et al., 2023b), a recent powerful human preference model trained on diverse high-quality human preference data, with winning rates up to 67% and 88%, respectively. Human preference, considered as the ground-truth ultimate evaluation metric for text-to-image generation, and objective evaluation metrics all generally support our methods.

## 2 PREREQUISITES

In this section, we formally introduce prerequisites and notations.

**Notations.** Suppose a diffusion model $\mathcal{M}$ can generate a clean sample $x_0$ based on some condition $c$, such as a general text prompt, given a sampled random noise $\epsilon$.[1] We denote the score neural network as $u_\theta(x_t, t)$, the model weights as $\theta$, the noisy sample at the $t$-th step as $x_t$, and $T$ as the total number of denoising steps.

**Diffusion Models.** The diffusion models (Ho et al., 2020) typically denoise a Gaussian noise along a reverse diffusion path (steps: $T \to 1$) to generate an image step by step. The probability via $u_\theta$, denoted as $p_\theta$, represents the sampling probability given the previous step's data. The starting point is sampled from a Gaussian distribution, $p(x_T) = \mathcal{N}(x_T | \mathbf{0}, \mathbf{I})$. The probability of the whole chain, $p_\theta(x_{0:T}) = p(x_T) \prod_{t=1}^{T} p_\theta(x_{t-1}|x_t)$. The deterministic sampling of $x_{t-1}$ in DDIM is as follows:

$$x_{t-1} = \sqrt{\overline{\alpha}_{t-1}} \left( \frac{x_t - \sqrt{1 - \overline{\alpha}_t} u_\theta(x_t, t)}{\sqrt{\overline{\alpha}_t}} \right) + \sqrt{1 - \overline{\alpha}_{t-1}} u_\theta(x_t, t), \tag{1}$$

where $a_t = 1 - \beta_t$ and $\overline{\alpha}_t = \prod_{t=1}^{s} \alpha_s$. The $\beta_t$ is the pre-defined parameters for scheduling the scales of adding noises. Based on the basic reverse process described above, many variations (Song et al., 2020; Sauer et al., 2023; Lin et al., 2024) have emerged.

**Noise Inversion.** The noise inversion is to invert a clean data into a noise along a pre-defined diffusion path. We can write the DDIM inversion process (Hertz et al., 2022) as

$$x_t \approx \sqrt{\frac{\alpha_t}{\alpha_{t-1}}} x_{t-1} + \sqrt{\alpha_{t-1}} \left( \sqrt{\frac{1 - \alpha_t}{\alpha_t}} - \sqrt{\frac{1 - \alpha_{t-1}}{\alpha_{t-1}}} \right) u_\theta(x_{t-1}, t, c), \tag{2}$$

where people approximate the denoising score prediction at $x_t$ with the inversion score prediction at $x_{t-1}$. We note that equation 1 can gradually transform a sampled noise $\epsilon$ into a generated sample $x_0$ along the denoising path, and equation 2 can gradually transform a generated sample $x_0$ back to a noise $\epsilon'$ along the noise inversion path. We note that the standard noising path which adds independent Gaussian noises is essentially different from the noise inversion path which adds the predicted noise of the score neural network $u_\theta$. While the generation denoising path and the noise inversion path are both guided by the score neural network $u_\theta$, the sampled noise $\epsilon$ and the inverse noise $\epsilon'$ are close but not identical due to the cumulative numerical differences.

**Fixed Points.** We denote the denoising-inversion transformation, $\epsilon \to x_0 \to \epsilon'$, as the transformation function $\epsilon' = F(\epsilon)$. If $\epsilon$ and $\epsilon'$ are ideally identical, namely $\epsilon = F(\epsilon)$, we call $\epsilon$ a fixed point of this mapping function $F$. In this case, the inverse noise $\epsilon'$ can perfectly recover the sample $x_0$ generated from $\epsilon$. This suggests that a state can remain fixed under some transformation. The fixed points have various great properties and many important applications in various fields, such as projective geometry (Coxeter, 1998), Nash Equilibrium (Nash Jr, 1950), and Phase Transition (Wilson, 1971).

## 3 METHODOLOGY

In this section, we first introduce the noise inversion stability hypothesis and show how it naturally leads to two novel noise-space algorithms, including noise selection and noise optimization.

---

**Algorithm 1** Noise Selection

1: **Input:** the diffusion model: $\mathcal{M}$, text prompt: $c$, the number of seeds: $K$
2: **Output:** the stable noise $\epsilon_s$
3: **for** $i = 1$ **to** $K$ **do**
4: seed $\leftarrow i$ // Set the random seed
5: Sampling a Gaussian noise $\epsilon_i$
6: $x_0 = \mathcal{M}(\epsilon_i, c)$
  // Generate an image
7: $\epsilon'_i = \text{Inversion}(x_0, c)$
  // Inverse noise
8: $s(\epsilon_i) = \cos(\epsilon_i, \epsilon'_i)$
9: **end for**
10: $\epsilon_s = \underset{\epsilon \in \{\epsilon_i | i=1,2,\cdots,K\}}{\arg\max}\ s(\epsilon)$
  // The noise with the highest stability score

---

**Algorithm 2** Noise Optimization

1: **Input:** the diffusion model: $\mathcal{M}$, text prompt: $c$, the number of gradient descent steps: $n$, the learning rate: $\eta$, the momentum value: $\beta$
2: **Output:** the optimized noise $\epsilon^\star$
3: Sampling a Gaussian noise $\epsilon$
4: **for** $i = 1$ **to** $n$ **do**
5: $x_0 = \mathcal{M}(\epsilon, c)$
  // Generate an image
6: $\epsilon' = \text{Inversion}(x_0, c)$
  // Inverse noise
7: $J(\epsilon) = 1 - \cos(\epsilon, \epsilon')$
8: $m_i = \beta m_{i-1} + \nabla_\epsilon J(\epsilon)$
9: $\epsilon = \epsilon - \eta m_i$
10: **end for**
  $\epsilon^\star = \epsilon$

---

[1] For simplicity, we abuse the latent space and the original data space in the presence of latent diffusion.

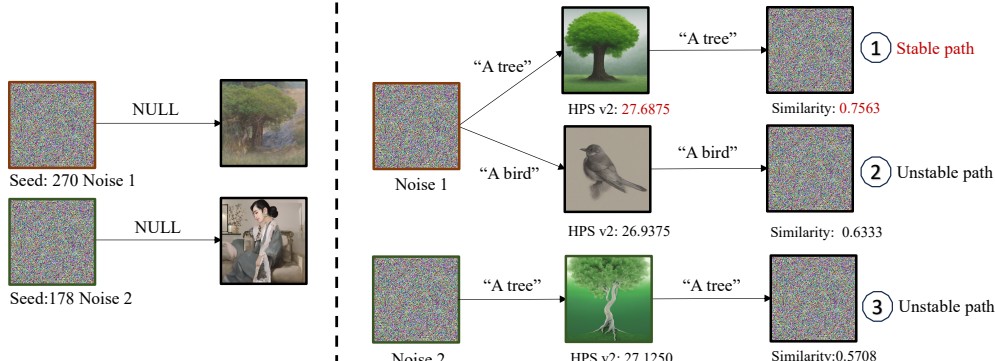

Figure 2: When the semantic information of noise and prompt are more similar, the noise is closer to the fixed point state under the denoising-inversion path. Left: the various semantic information implicit in different initial noises. We pick the prompt ("A tree") related to noise1 and the unrelated prompt ("A bird"). Right: the stronger the correlation between the noise and the prompt, the better the result, with greater similarity between the initial noise and the inverse noise.

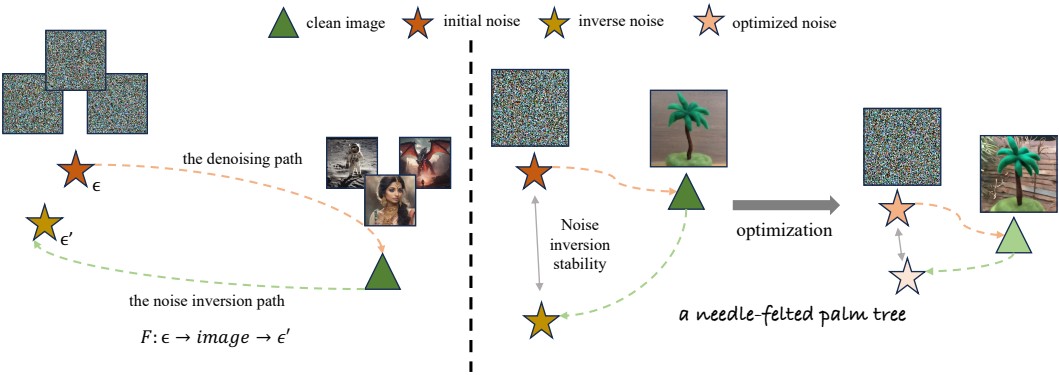

Figure 3: The overview of noise inversion stability and noise optimization. Left: the denoising-inversion path. If $\epsilon$ is a fixed-point noise, then $\epsilon = \epsilon'$. Right: random noises are not perfect fixed-point noises for the denoising-and-inversion transformation, which leads to some difference between original noises and inverse noises. We can select stable noises or directly optimize the given random noises to get closer to a fixed point.

**Noise Inversion Stability.** It is well known that fixed points are stable under the transformation and, thus, have great properties (Burton, 2003; Connell, 1959; Pata et al., 2019). May the fixed-point Gaussian noises under the denoising-inversion transformation $F$ also exhibit some advantages? As finding the fixed points of this complex dynamical system is intractable, unfortunately, we cannot empirically verify it. Instead, we can formulate Definition 1 to measure the stability of noise for the denoising-inversion transformation $F$.

**Definition 1** (Noise Inversion Stability). *We define $F$ as the denoising-inversion transformation, $\epsilon \to x_0 \to \epsilon'$. Suppose a sampled noise $\epsilon$ has its inverse noise $\epsilon' = F(\epsilon)$ given by a diffusion model $\mathcal{M}$ with the condition $c$. We define the noise inversion stability of the sampled noise $\epsilon$ as*

$$s(\epsilon) = \cos(\epsilon, \epsilon') \tag{3}$$

*for the diffusion $\mathcal{M}$ with the condition $c$, where $\cos$ is the cosine similarity between two vectors.*

We use cosine similarity to measure stability for simplicity, while it is also possible to use other similarity metrics. The results using other metrics can be found in Appendix B. Our empirical analysis in Section 4 suggests that the simple cosine similarity metric works well.

**Evidence.** We refer to the content generated under the "NULL" prompt as the semantic information implicit in the corresponding initial noise (see the left part of Figure 2). This semantic information reflects the generation trends of sampling noise points. For instance, noise1 tends to produce a "tree" layout of displays. We calculate the inversion stability score for three scenarios: (1) noise and prompt matching (setting ① in Figure 2), (2) noise and prompt not matching (settings ② and ③ in Figure 2). We observed a strong correlation between the inversion stability scores and the degree of match, with higher inversion stability scores leading to better image quality. When the noise matches the prompt, the noise is closer to the fixed point in the denoising-inversion path guided by the prompt.

**Noise Selection.** Based on the evidence results and inspired by the intriguing mathematical properties of fixed points, we hypothesize that the noise with higher inversion stability can lead to better results. If this hypothesis is reasonable, this naturally provides a novel and useful noise selection algorithm that selects the noise seed with the highest stability score from $K$ noise seeds (e.g. $K = 100$ in this work). We present the pseudocode in Algorithm 1.

**Noise Optimization.** As we have an objective to increase the noise inversion stability, is it possible to actively optimize a given noise by maximizing the stability score? We further propose the noise optimization algorithm that directly performs Gradient Descent (GD) on the loss, $1 - \cos(\epsilon, \epsilon')$, with respect to $\epsilon$, where we keep the diffusion model weights and $\epsilon'$ constant for each optimization step. We take the diffusion models as a fixed mapping function and the optimization objective is directly act on the noise. This make us directly optimize the initial noise and do not let gradients flow through the network, greatly saving the memory. We present the illustration of noise optimization in the right column of Figure 3. We present the pseudocode in Algorithm 2.

## 4 EMPIRICAL ANALYSIS

In this section, we conduct extensive experiments to demonstrate the effectiveness of our methods. We take text-to-image generation as our main setting.

### 4.1 EXPERIMENTAL SETTINGS

**Models:** SDXL-turbo (Sauer et al., 2023) and SDXL (Podell et al., 2023). SDXL is a representative and powerful diffusion model. SDXL-turbo is a recent accelerated diffusion model that can produce results better than standard SDXL but only takes 4 denoising steps. We choose the denoising steps for SDXL-turbo as 4 steps and SDXL as 10 steps for reducing computational time and carbon emissions, unless we specify otherwise. We use the model's default scheduler for denoising and DDIM scheduler for inversion in experiments. We also empirically study the impact of denoising steps on the optimization effectiveness in Section 4.3.

**Dataset:** We select common datasets to evaluate our algorithm's performance in the general text-to-image task. We use all 200 test prompts from the DrawBench dataset (Saharia et al., 2022) which contain comprehensive and diverse descriptions beyond the scope of the common training data. We use the first 100 test prompts from the Pick-a-Pic (Kirstain et al., 2024) which consist of interesting prompts gathered from the users of the Pick-a-Pic web application. We also use HPD v2 dataset which contains 400 prompts and related results are shown in Appendix B.

**Evaluation metrics:** We evaluate the quality of the generated images using both human preference and popular objective evaluation metrics, including HPS v2 (Wu et al., 2023b), AES (Schuhmann et al., 2022), PickScore (Kirstain et al., 2024), and ImageReward (Xu et al., 2024). AES indicates a conventional aesthetic score for images, while HPS v2, PickScore, and ImageReward are all emerging human reward models that approximate human preference for text-to-image generation. Particularly, HPS v2 is a better human reward model and offers a metric closer to human preference (see Table 6 in (Wu et al., 2023b)) than other objective evaluation metrics. Moreover, human preference is regarded as the ground truth and ultimate evaluation method for text-to-image generation. Thus, we regard human preference and HPS v2 as the two most important metrics.

**Hyperparameters:** For the noise selection experiments, we select the (most) stable noise and the (most) unstable noise from 100 noise seeds according to the noise inversion stability. We evaluate generated results using human preference and objective evaluation metrics. For the noise optimization experiments, we initialize the noise using one random seed and perform GD to optimize the noise with

Table 1: The quantitative results of noise selection. Each reported score is the mean score over all evaluated prompts. The corresponding winning rate results are shown in Figure 4 and the qualitative results are shown in Figure 5. Model: SDXL-turbo.

| Dataset | Noise | HPS v2 | AES | PickScore | ImageReward | Average |
|---------|-------|--------|-----|-----------|-------------|---------|
| Pick-a-Pic | Unstable noise | 27.2688 | 5.9265 | 21.6227 | 0.7812 | 13.8998 |
| | Stable noise | **27.4934** | **5.9960** | **21.6372** | **0.8981** | **14.0062** |
| DrawBench | Unstable noise | 28.1377 | 5.3945 | **22.4251** | 0.7021 | 14.1646 |
| | Stable noise | **28.4266** | **5.6082** | 22.4200 | **0.7325** | **14.2968** |
| HPD v2 | Unstable Noise | 28.3594 | 5.9663 | 22.4641 | 0.9525 | 14.4356 |
| | Stable Noise | **28.6250** | **6.0075** | **22.4644** | **0.9856** | **14.5206** |

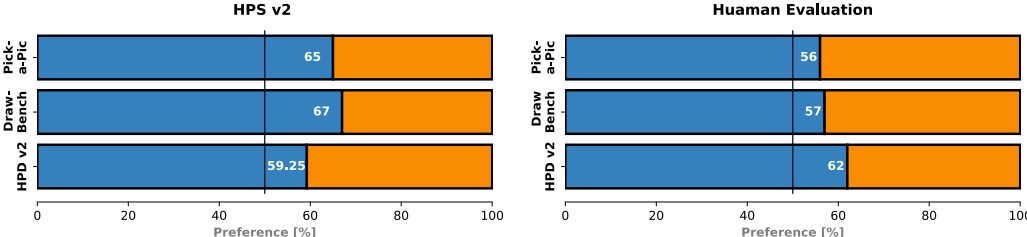

Figure 4: The winning rate results from noise selection. The blue bars represent the side of stable noises. The orange bars represent the side of unstable noises. Mode: SDXL-turbo.

100 steps. The default values of the learning rate and the momentum are 100 and 0.5, respectively. More details can be found in Appendix A.2.

## 4.2 THE EXPERIMENTS OF NOISE SELECTION

The noise selection experiments are to compare the results denoised from stable noises and unstable noises, where the noise with the highest stability score is the stable noise, and the noise with the lowest stability score is defined as unstable noise.

**Quantitative results.** We present the objective evaluation scores in Table 1. The HPS v2 is the main objective evaluation metric. The HPS v2 score of stable noises surpasses its counterpart of unstable noise by 0.225, 0.289 and 0.266, respectively, on Pick-a-Pic, DrawBench and HPD v2. The average scores also support the advantage of stable noises over unstable noises. The quantitative results support the noise inversion stability hypothesis and suggest that stable noises often significantly outperform unstable noises in practice.

Besides the scores, the winning rates can tell the percentage of one result better than the other on the evaluated prompts. We particularly show the winning rates of human preference and HPS v2 in Figure 4 to visualize two representative evaluation metrics. All winning rates are significantly higher than 50%. The human preference winning rates are up to **56%**, **57%** and **62%**, respectively, over Pick-a-Pic, DrawBench and HPD v2, while the HPS v2 winning rates are even up to **65%**, **67%** and **59.25%** respectively.

**Qualitative results.** We conduct case studies for qualitative comparison. We not only care about the standard visual quality, but also further focus on those challenging cases for diffusion models, such as color, style, text rendering, object co-occurrence, position, and counting. The results in Figure 5 show that the images denoised from stable noise are significantly better than images denoised from unstable noise in various aspects. 1) Color: the stable noise leads to a yellow fork accurately, while the unstable noise can only lead to a yellow hand with an incorrect fork. 2) Style: the stable noise obviously corresponds to the "1950s batman comic" style more precisely with rich background details. 3) Text rendering: the stable noise can render the correct "diffusion". 4) Object co-occurrence,

Figure 5: The qualitative results of noise selection. The results highlight the improvements of stable noises in various aspects, such as color, style, text rendering, object co-occurrence, position, and counting. The prompts are from the benchmark datasets. Model: SDXL-turbo.

Table 2: The quantitative results of noise optimization. The qualitative results are shown in Figure 7, and the winning rate results are shown in Figure 6. Model: SDXL.

| Dataset | Noise | HPS v2 | AES | PickScore | ImageReward | Average |
|---------|-------|--------|-----|-----------|-------------|---------|
| Pick-a-Pic | Original Noise | 25.9800 | 5.9903 | 21.0183 | 0.2500 | 13.3207 |
| | Optimized Noise | **26.6422** | **6.0504** | **21.2344** | **0.4622** | **13.5973** |
| DrawBech | Original Noise | 26.6203 | 5.4889 | 21.4815 | 0.0575 | 13.4121 |
| | Optimized Noise | **27.3651** | **5.5438** | **21.6508** | **0.1767** | **13.6841** |
| HPD v2 | Random Noise | 26.8750 | 6.0185 | 21.8770 | 0.4597 | 13.8076 |
| | Optimized Noise | **27.8125** | **6.0722** | **22.0395** | **0.6449** | **14.1423** |

the stable noise can generate correct combinations of two objects, while the unstable noise falsely merges two concepts together. 5) Position, the stable noise corrects the wrong position relation of the unstable noises. 6) Counting, the stable noises accurately correct the number of both cats and dogs.

In summary, both quantitative and qualitative results demonstrate the significant effectiveness of noise selection according to the noise inversion stability.

## 4.3 THE EXPERIMENTS OF NOISE OPTIMIZATION

The noise optimization experiments are to compare the results of original noises and optimized noises. For each prompt, we sample a Gaussian noise as the original noise and learn optimized noises by Algorithm 2. Note that optimized noises are approximately but not real Gaussian noises.

**Quantitative results.** We present quantitative results in Table 2. All objective evaluation metrics in the experiment consistently support the advantage of optimized noises over original noises. The HPS v2 score of optimized noises surpasses its counterpart of original noises by 0.662, 0.745 and 0.938, respectively, on the Pick-a-Pic, DrawBench, and HPD v2. The average score again supports the advantage of optimized noises over original noises.

Similarly, we visualize the winning rates of the two most important metrics, HPSv2 and human preference to show the percentage of improved cases in Figure 4. The human preference winning rates of noise optimization are **69%**, **72.5%** and **83.25%**, respectively, over Pick-a-Pic, DrawBench and HPD v2, while the HPS v2 winning rates are even up to **87%**,**88%** and **87.25%**. The winning rate improvements are comparable to the performance gap between two generations of SD models, such as SDXL-turbo (Sauer et al., 2023) and cascaded pixel diffusion models (IF-XL) (Saharia et al., 2022). We also compare our method with DOODL in Appendix B.

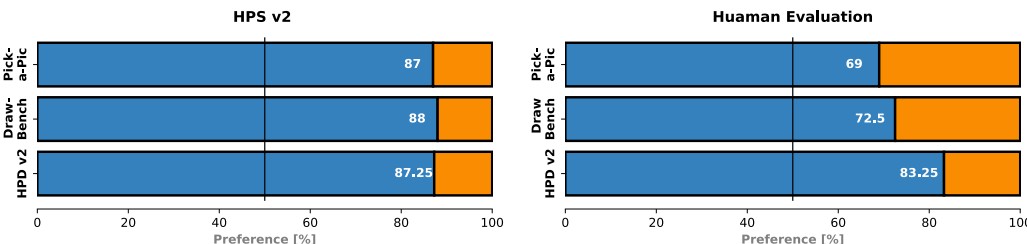

Figure 6: The winning rate result of noise optimization. The blue bars represent the side of optimized noises. The orange bars represent the side of the original noise. Model: SDXL.

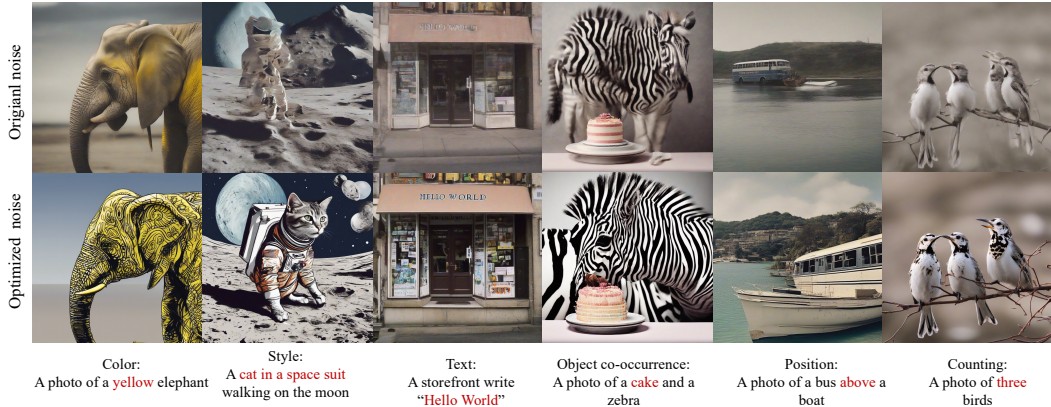

Figure 7: The qualitative results of noise optimization on benchmark datasets. Each pair of results is generated by SDXL. The results demonstrate that the optimized noise outperforms the original noise in various aspects, such as color, style, text rendering, object co-occurrence, position, and counting.

**Qualitative results.** We present the qualitative results of original noises and optimized noises in Figure 7. Similar to what we observe for noise selection, noise optimization also improves multiple challenging cases, such as color, style, text rendering, object co-occurrence, position, and counting we mentioned above. Moreover, we also present examples that noise optimization can improve the details of human characters and bodies in Figure 8. Optimized noises can lead to more accurate human motion and appearance. For example, the huntress's hand generated by the optimized noise is accurately holding the end of the arrow.

**Impacts of denoising steps $T$ on optimization.** The noise inversion process directly depends on the number of denoising steps $T$. We apply our noise optimization to SDXL with various denoising steps to study the impact of denoising steps. We present the winning rates of noise selection with $T \in \{5, 10, 30, 50\}$ in Figure 9. The results show that the improvement of noise optimization is relatively robust to a wide choice of denoising steps. Optimized noises are especially good for very few denoising steps.

**Noise Optimization for 3D Generation.** It is easy to see that the proposed methods can be generally applied to other diffusion models. Here, we provide an example. We apply noise optimization to 3D generation tasks with a popular image-to-3D generative model, SV3D (Voleti et al., 2024). We clearly observe the improvements in the body details of these 3D characters. Due to the space limit, we leave more experimental details and results in Appendix E.

In summary, noise optimization can significantly improve generated results in multiple challenging aspects. It is especially surprising that optimized noises deviated from Gaussian noises can help diffusion models generate better results than real Gaussian noises.

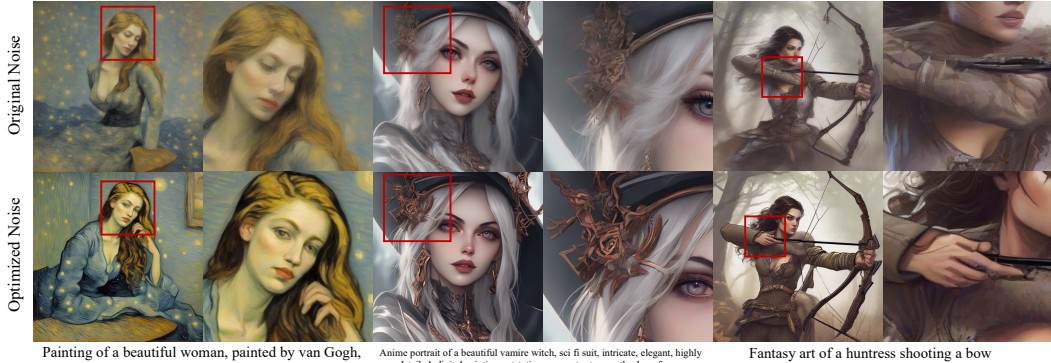

Painting of a beautiful woman, painted by van Gogh, starry night

Anime portrait of a beautiful vamire witch, sci fi suit, intricate, elegant, highly detailed, digital painting, artstation, concept art, smooth, sharp focus, illustration, art by grep rutkowski

Fantasy art of a huntress shooting a bow

Figure 8: The character and body details of original noises and optimized noises. The prompts are from Pick-a-Pic. Model: SDXL

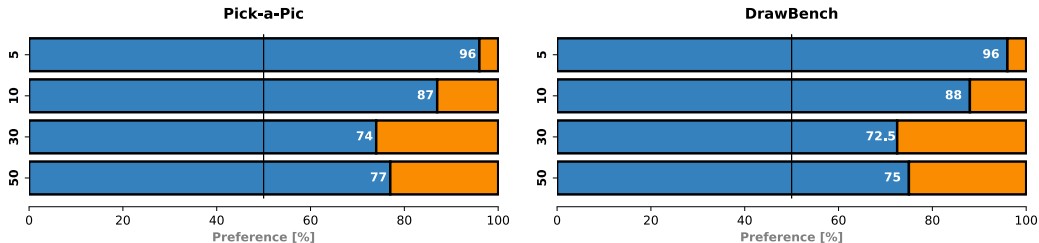

Figure 9: The winning rates of noise optimization with respect to various denoising steps. Metric: HPS v2. Model: SDXL.

## 4.4 SUPPLEMENTARY RESULTS

We provide details of our experimental settings and additional results in the Appendix. (1) Experimental settings: Appendix A presents the experimental settings and hyperparameter details. (2) Ablation and comparison experiments: Appendix B presents results using different similarity measures(Tables 3 and 5), and comparisons with other methods(Table 6). (3) Related work and discussion: Appendix C presents the detailed contents. (4) Additional task results: Appendix E presents results for the image-to-3d task(Table 8).

## 5 DISCUSSION AND LIMITATIONS

In this section, we discuss related works and three main limitations of our method.

**Related work.** Some works (Wallace et al., 2023; Chefer et al., 2023; Agarwal et al., 2023; Guo et al., 2024) have noted that noise plays a significant role in the final results. However, they typically require additional information to optimize the noise, such as image quality evaluators or token IDs used for constructing attention loss. The first type of methods relies on the performance of the evaluator and demands more memory due to the need for gradients through the denoising process. The second type operates in the noise space but requires user-specified token IDs or an LLM to extract them, which limits their applicability to general prompts. In contrast, our method is based on a mathematical property of noise, allowing it to work directly within the noise space and adapt to general prompts. In summary, from a perspective of real-world practice, previous so-called noise optimization methods cannot be directly applied or compared with our method for a general text-to-image generation task. We discuss more about this key point in Appendix C.

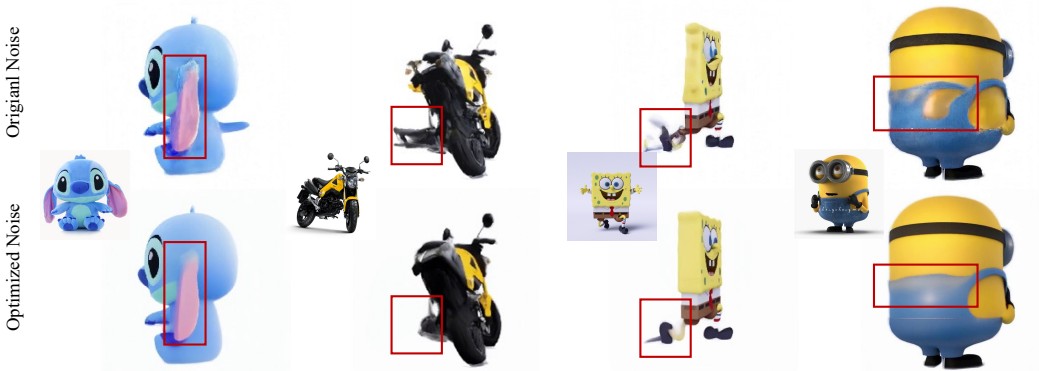

Figure 10: The qualitative results of noise optimization for 3D generation. The small images are the input. The red box highlights the differences. Model: SV3D.

**Theoretical Understanding.** With the inspirations from fixed points in dynamic systems, we still do not theoretically understand why not all noises are created for diffusion models. We formulated and empirically verified the hypothesis that random noises with higher inversion stability often lead to better results, it is still difficult to theoretically analyze how the performance of diffusion models mathematically depends on noise stability. We believe theoretically understanding noise selection and noise optimization will be a key step to further improve them.

**Optimization Strategies.** In this work, we only applied simple gradient descent with multiple (e.g., 100) steps to optimize the noise-space loss, but noise optimization seems like a difficult optimization task. In some cases, we observe that the loss does not converge smoothly. Due to computational costs and a limited understanding of the noise-space loss landscape, we did not carefully fine-tune the hyperparameters or employ advanced optimizers, such as Adam (Kingma & Ba, 2015) in this work. Thus, while the current optimization strategy works well, it is far from releasing the power of noise-space algorithms. We think it will be very promising and important to better analyze and solve this emerging optimization task with advanced optimization methods.

**Computational Costs.** Both noise selection and noise optimization require significantly more computational resources and time compared to standard generation. For noise selection, we compute the inversion stability loss across 100 noise seeds and select the one with the highest stability score, repeating the forward and inversion processes 100 times. In noise optimization, we perform gradient descent over 100 steps, repeating the forward pass and inversion 100 times. While accelerating noise selection may be challenging, noise optimization could likely be sped up by reducing the number of gradient descent steps in the future.

## 6 CONCLUSION

In this paper, we report an interesting noise inversion stability hypothesis and empirically observe that noises with higher inversion stability often lead to better results. This hypothesis motivates us to design two novel noise-space algorithms, noise selection and noise optimization, for diffusion models. To the best of our knowledge, we are the first to apply the selection and optimization methods in a pure noise space that does not involve any additional estimators and extra annotated information. Unlike previous related methods that require specifically designed prompts, both of our algorithms can directly adopt to general text-to-image generation with standard text prompts. Our extensive experiments demonstrate that the proposed methods can significantly improve multiple aspects of qualitative results and enhance human preference rates as well as objective evaluation scores. Moreover, the proposed methods can be generally applied to various diffusion models in a plug-and-play manner. While some limitations exist, our work has made the first solid step to explore this promising direction. We believe our work will motivate more studies on understanding and improving diffusion models from the perspective of noise space.

ETHICS STATEMENT

Our work proposes noise selection and optimization for diffusion models. As previously emphasized, our algorithm does not introduce additional information, ensuring that the generated results remain free from any ethical biases. We have also confirmed that none of the data used in our experiments presents ethical risks. Given the potential impact of our algorithms in both academic and commercial settings, we stress the importance of responsible use to minimize risks such as misuse or unintended harm.

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

## A  EXPERIMENTAL SETTINGS OF MAIN EXPERIMENTS

**Computational environment.** The experiments are conducted on a computing cluster with GPUs of NVIDIA® Tesla™ A100.

### A.1  DATASETS AND DATA PREPROCESSING

we conduct experiments across three datasets as follows:

**Pick-a-Pic.** (Kirstain et al., 2024): This dataset is composed of data collected from users of the Pick-a-Pic web application. Each example in this dataset consists of a text prompt, a pair of images, and a label indicating the preferred image. It is worth noting that for fast validation and saving computational resources, we only use the first 100 prompts as text conditions to generate images in the main experiment.

**DrawBench.** (Saharia et al., 2022): The examples in this dataset contain a prompt, a pair of images, and two labels for visual quality and prompt alignment. The total number of examples in this dataset is approximately 200. This dataset contains 11 categories of prompts that can be used to test various properties of generated images, such as color, number of objects, text in the scene, etc. The prompts also contain long, complex descriptions, rare words, etc.

**HPD v2 Wu et al. (2023b):** HPD v2 comprises a test split and a training split. The test split consists of 400 groups of images. Among them, 300 groups use prompts from DiffusionDB Wang et al. (2022) , and 100 groups use prompts from COCO Captions Lin et al. (2014).

**The difference between these datasets**: The prompts in Pick-a-Pic are from real users and have more daily descriptions. The prompts in DrawBench have more complex descriptions and contain rare words. The prompts in HPD v2 contain more comprehensive situation.

In all main experiments, we set all tensors as half-precision to improve experimental efficiency. In calculating the inversion stability, we expand the noise tensor to a one-dimensional vector along the channel dimension.

### A.2  THE HYPERPARAMETERS

**Noise Selection.** In noise selection experiments, for each prompt, we sample 100 noises using random seeds from 0 to 99. According to the inversion stability score, we select the stable noise among all candidate noises, using the algorithm 1.

**Noise Optimization.** In noise optimization experiments, for each prompt, we first randomly sample a noise using a random seed selected from 0 to 99. This noise is denoted as the original noise. We use algorithm 2 to optimize the original noise with 100 gradient descent steps. We set the defaulted learning rate is 100 and equip it the learning rate with a cosine annealing schedule. The default value of momentum is 0.5.

**Hyperparameter tuning and fluctuation.** We need to choose a relatively large learning rate, e.g. 100, which matters to successful optimization, as the order of magnitude of the gradient norm is about $10^{-5}$. In our experiment, we select the optimal learning rate $\eta$ from {0.1, 1, 10, 100, 1000}. Moreover, we chose the momentum value as 0.5 with careful fine-tuning, as the momentum value does not significantly affect the final results. According to the empirical analysis, the performance closely depends on the learning rate due to the convergence problem and is robust to other hyperparameters. This is not strange, as the learning rate matters to nearly all optimization tasks.

### A.3  EVALUATION METRICS

**Human Preference Score v2 (HPS v2)**: This score is calculated by a finetuned CLIP[2] on the HPD v2 dataset (Wu et al., 2023b), a comprehensive human preference dataset. This human preference dataset is known for its diversity and representativeness. Each instance in the dataset contains a pair of images with prompts and a label of human preference.

---

[2]The CLIP version is ViT-H/14

**Aesthetic Score (AES)**: The AES[3] is calculated by the Aesthetic Score Predictor (Schuhmann et al., 2022), which is designed by adding five MLP layers on top of a frozen CLIP[4] and only the MLP layers are fine-tuned by a regression loss term on SAC (Pressman et al., 2022), LAION-Logos[5] and AVA (Murray et al., 2012) datasets. The score ranges from 0 to 10. A higher score means the image has better visual quality.

**PickScore**: This is also involves a human preference model, where the score is generated by a fine-tuned CLIP model. This model has been trained on the Pick-a-Pic dataset, which contains a large number of user-annotated samples reflecting human preferences.

**ImageReward**: This is an early human preference model (Xu et al., 2024).

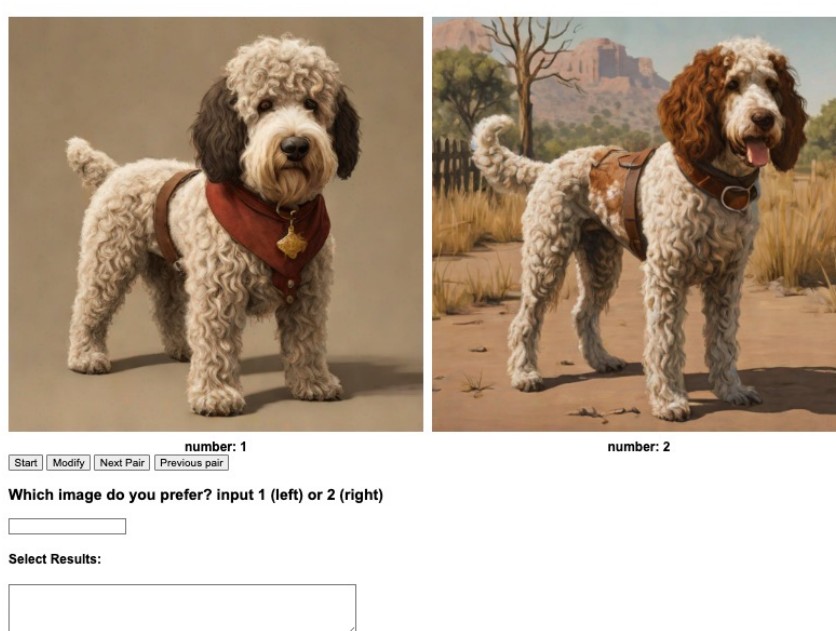

Figure 11: The web page for human evaluation.

---

[3]The Github page: https://github.com/christophschuhmann/improved-aesthetic-predictor
[4]The CLIP version is ViT-H/14
[5]https://laion.ai/blog/laion-aesthetics/

**Human evaluation**: Human annotators select a better one from a pair of images following the criteria:

- The correctness of semantic alignment

- The correctness of object appearance and structure

- The richness of details

- The aesthetic appeal of the image

- Your preference for upvoting or sharing it on social networks

We built a web page for human evaluation, as shown in Figure 11.

**The difference between these metrics**: The AES is primary for evaluating the visual quality, while others are for human preference and text-image alignment.

# B  SUPPLEMENTARY EXPERIMENTAL RESULTS

## B.1  THE EXPERIMENTS OF OTHER SIMILARITY METRICS

We present the comparative results of various metrics here following the setting of Table 1 in the main paper. We show the winning rate results of each metric on Pick-a-Pic, DrawBench and HPD v2 datasets.

Table 3: The winning rate results of noise selction. Model: SDXL-turbo. Dataset: Pick-a-Pic.

| Metrics | HPS v2 | AES | PickScore | ImageReward | Avg. rate |
|---|---|---|---|---|---|
| Cosine similariy | 65% | 54% | 51% | 55% | 56.25% |
| MSE | 54% | 52% | 50% | 51% | 51.75% |
| MAE | 55% | 50% | 52% | 54% | 52.5% |

Table 4: The winning rate results of noise selection. Model: SDXL-turbo. Dataset: DrawBench.

| Metrics | HPS v2 | AES | PickScore | ImageReward | Avg. rate |
|---|---|---|---|---|---|
| Cosine similariy | 67% | 58% | 48.5% | 57% | 57.63% |
| MSE | 68.5% | 53% | 56.5% | 55% | 58.25% |
| MAE | 65% | 53% | 51% | 56% | 56.25% |

Table 5: The winning rate results of noise selection. Model: SDXL-turbo. Dataset: HPD v2.

| Metrics | HPS v2 | AES | PickScore | ImageReward | Avg. rate |
|---|---|---|---|---|---|
| Cosine similariy | 61.75% | 55.75% | 50.75% | 49.50% | 54.44% |
| MSE | 59% | 51.25% | 53% | 51.50% | 53.69% |
| MAE | 57.25% | 51.75% | 49.50% | 56% | 53.63% |

The results show that the cosine similarity metric has good performance on both datasets compared to the Mean Squared Errors (MSE) and Mean Absolute Errors (MAE). This suggests that the cosine similarity is a more effective evaluation metric, further justifying its use in noise selection and optimization and optimization experiments.

## B.2 THE COMPARISON EXPERIMENTS

We select DOODL (Wallace et al., 2023) as a comparison method. DOODL leverages CLIP to guide noise optimization for improved results, but it requires additional memory to backpropagate gradients through the diffusion model (which indirectly affects the noise), and its performance is influenced by the evaluator's choice. In contrast, our method is mathematically grounded, operates entirely within the noise space, and does not rely on an external evaluator, thus avoiding additional bias. While DOODL uses SD 1.4 by default, our method, with similar memory usage, supports SDXL. For a fair comparison, we report the winning rates to highlight the improvements achieved through noise optimization. The experiments are conducted on the Pick-a-Pic, DrawBench, and HPD v2 datasets.

Table 6: The winning rate results of DOODL (Wallace et al., 2023) and ours.

| Dataset | Method | HPS v2 | AES | PickScore | ImageReward | Ave. rate |
|---------|--------|--------|-----|-----------|-------------|-----------|
| Pick-a-Pic | DOODL | 69.00% | 50.00% | 67.00% | 62.00% | 62.00% |
| | Ours | 83.00% | 55.00% | 67.00% | 68.00% | 68.25% |
| DrawBench | DOODL | 52.5% | 51.00% | 62% | 59.5% | 56.25% |
| | Ours | 87.00% | 55.00% | 64.00% | 68.00% | 68.50% |
| HPD v2 | DOODL | 59.50% | 50.00% | 61.25% | 61.00% | 57.94% |
| | Ours | 87.25% | 57.75% | 63.50% | 71.50% | 70.00% |

The results demonstrate that our method outperforms DOODL across all metrics on the three datasets. This consistent improvement stems from the robustness of our approach, which is grounded in mathematical principles, making it adaptable to various data types. And, our method is not limited by the performance of the evaluation model, allowing for more reliable results.

## C RELATED WORK AND DISCUSSION

**Noise inversion and editing**. The noise inversion technique is mainly applied in image editing (Mokady et al., 2023; Meiri et al., 2023; Huberman-Spiegelglas et al., 2023) in very similar ways. They usually invert a clean image into a relatively noisy one via a few inverse steps and then denoise the inverse noisy images with another prompt to achieve instruction editing. Some work (Mao et al., 2023) in this line of research realized that editing noises can help editing generated results. Specifically, modifying a portion of the initial noise can affect the layout of the generated images. Other works (Liu et al., 2024; Shi et al., 2023) focused on dragging and dropping image content via interactive noise editing. However, the goal of previous studies is to control image layout under fine-grained control conditions, such as input layout or editing operations. In contrast, we focus on generally improving the generated results of diffusion models by selecting or optimizing a Gaussian noise according to the stability score.

**Noise selection and optimization**. In recent years, there have been some works on selecting/optimizing the results in the noise space. Typically, these methods rely on additional information, such as image quality evaluators or token IDs provided by the user/extracted by a large language model (LLM) to construct a "noise-prompt" attention loss. However, our method is based on a mathematical property and directly operates on the pure noise space, adopting for general prompts and various models.

(1) For example, Karthik et al. (2023) first generates many candidate images and selects the best one from these by comprehensive scoring from a VAQ model (e.g. GPT) and an image quality evaluator (e.g. ImageReward). (Selecting images equals selecting the initial noises). This type of selecting method is a post-selection method that cannot directly judge the quality of the noise and is seriously affected by the quality of the evaluator. The potential for introducing additional bias may also increase. In contrast, our method evaluates the initial noise via inverse stability. This is a mathematical property that is independent of any additional input information and does not introduce additional bias.

(2) Some noise optimization methods mainly target image quality scores to optimize noise values. For example, DOODL (Wallace et al., 2023) takes the score from an additional image quality evaluator

as the optimization target to gradually change the values of initial noise. However, it depends on the performance of the chosen image quality evaluator, which increases the likelihood of introducing additional bias and memory usage. Moreover, the evaluation scores require gradient backpropagation to influence the initial noise, resulting in significantly higher memory usage. Other methods use the attention score map and user-specified or LLM extraction token IDs to design a "noise-prompt" attention loss to maximum correlation between noise features and special tokens. The general form of the loss function is as follows:

$$loss = 1 - \min_{y_i \in \mathcal{Y}} \max(\mathcal{A}_{y_i}), \tag{4}$$

where $\mathcal{Y}$ denotes the set of target tokens, $A_{y_i}$ denotes the attention map that corresponds to the $y_i$ token. A&E (Chefer et al., 2023) and A-STAR (Agarwal et al., 2023) apply attention loss at each denoising step, which may lead to over-optimization or under-optimization. INITNO (Guo et al., 2024) sets a threshold and applies attention loss to the initial noise and each step noise. However, relying in the attention loss can only optimize results for selected tokens, limiting these methods to work with concept combination prompts (e.g., *A+B* prompts like "a cat and a dog"). They struggle to handle high-level concepts such as *style*, *detail descriptions*, and similar abstract elements.

In contrast, our method directly evaluates and optimizes the initial noise, reducing memory usage and accommodating general prompts. Table 7 shows more comparisons between above methods and ours. For a reasonable comparison, we only compare with DOODL in Appendix B.2.

| Methods | SD version | Prompt type | Optimization object | Extra information |
|---------|-----------|-------------|---------------------|-------------------|
| DOODL | SD 1.4 | General | Initial noise | CLIP |
| A&E | SD 1.4/1.5 | A+B | Each step noise | Token IDs |
| A-STAR | - | A+B | Each step noise | Token IDs |
| INITNO | SD 1.4/2.1 | A+B | Initial noise & each step noise | Token IDs |
| Ours | SDXL | General | Initial noise | None |

Table 7: The comparison of other noise optimization methods and Ours

## D  SUPPLEMENTARY EXPERIMENTAL RESULTS

We show more results of noise optimization experiments in Figure 12

## E  3D OBJECT GENERATION

In this section, we analyze noise optimization for 3D diffusion models.

### E.1  METHODOLOGY

The noise inversion rule of image-to-3D diffusion models is different from text-to-image diffusion. Here we derive the noise inversion rule for the popular image-to-3D diffusion model, SV3D (Voleti et al., 2024).

SV3D employs the EDM framework (Karras et al., 2022), which improves upon DDIM with a reparameterized to the denoising process. Taking a single image as input, SV3D generates a multi-view consistent video sequence of the object based on a specified camera trajectory, showcasing remarkable spatio-temporal properties and generalization capabilities. Specifically, we choose the SV3D-U variant, which, during training, consistently conditions on a static trajectory to generate a 21-frame 3D video sequence, with each frame representing a 360/21-degree rotation of the object.

The denoising process $x_{t+1} \to x_t$ within the EDM framework can be written as

$$x_t = x_{t+1} + \frac{\sigma_t - \sigma_{t+1}}{\sigma_{t+1}} \mu, \tag{5}$$

$$\mu = x_{t+1} - \left( c_{skip}^{t+1} x_{t+1} + c_{cout}^{t+1} u_\theta(c_{in}^{t+1} \hat{x}_{t+1}; c_{noise}^{t+1}) \right). \tag{6}$$

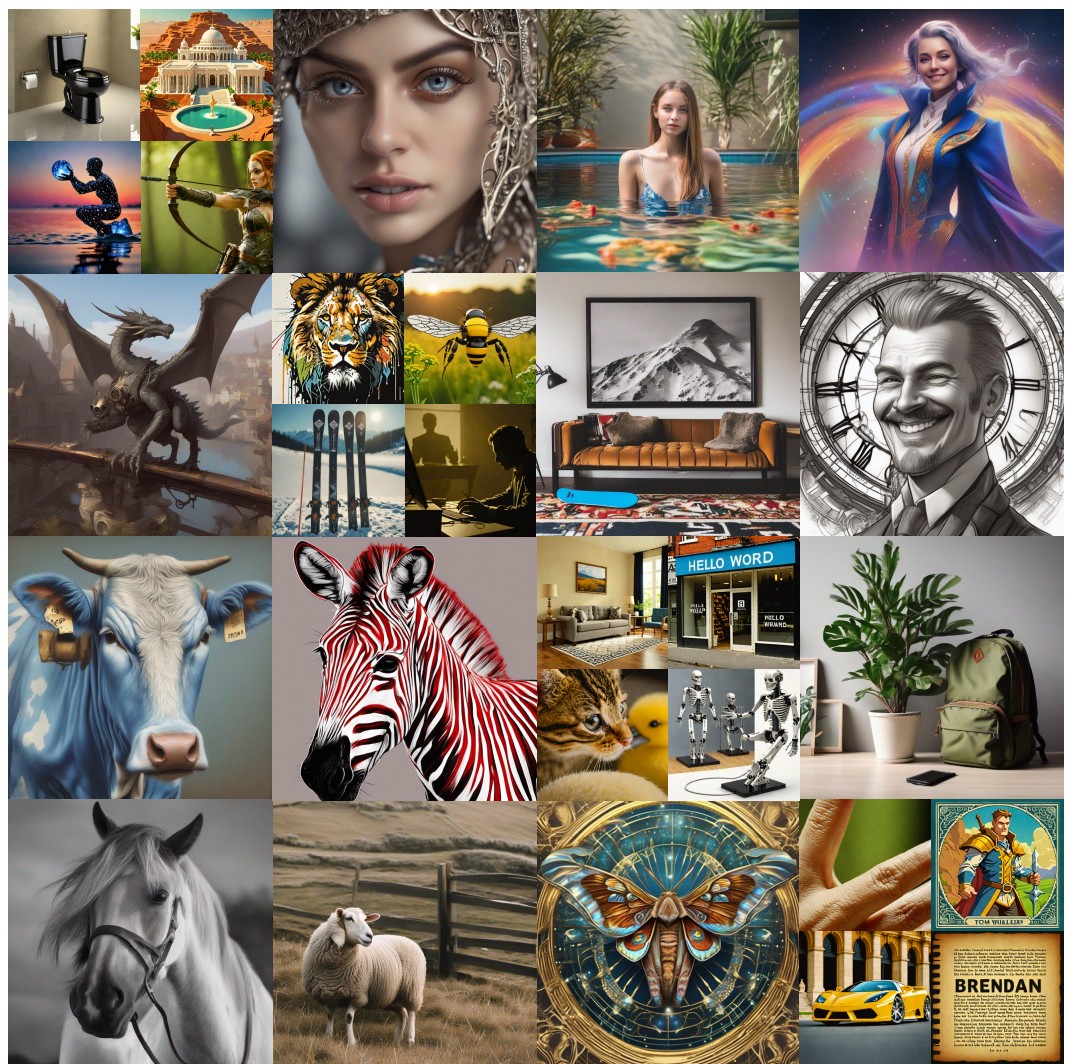

Figure 12: More results of optimized noises. The large images are generated by SDXL, and small images are generated by SDXL-turbo.

We denote $\sigma_t$ as the noise level of the scheduler at the $t$-th time step and $u_\theta$ denotes the scoring network. $c_{skip}$, $c_{out}$, $c_{in}$, and $c_{noise}$ are coefficients dependent on the noise schedule and the current time step $t$ in the Euler sampling method. Subsequently, if we intend to achieve noise inversion $\hat{x}_t \to \hat{x}_{t+1}$, we can modify Equation $equation$ 5 accordingly as

$$\hat{x}_{t+1} = \frac{\sigma_{t+1}\hat{x}_t + (\sigma_t - \sigma_{t+1})\, c_{out}^{t+1} u_\theta\left((c_{in}^{t+1}\hat{x}_{t+1}; c_{noise}^{t+1}\right)}{(\sigma_t - \sigma_{t+1})\left(1 - c_{skip}^{t+1}\right) + \sigma_{t+1}}. \tag{7}$$

Following previous work (Fan et al., 2024; Hertz et al., 2022), during the noise inversion process, we have utilized the noise prediction results at $x_t$ to approximate those at $x_{t+1}$.

## E.2 EXPERIMENTAL SETTING

### E.2.1 DATASETS

We randomly sample 30 objects from the OmniObject3D Dataset (Wu et al., 2023a) and render them using Blender's Eevee engine. Each object is rendered in a video sequence comprising 84 frames,

with the camera rotating 360/84 degrees between each frame. Additionally, we set the ambient lighting to a white background to match the conditions stipulated by SV3D. It is important to note that, as SV3D has not disclosed the rendering details of its test dataset, achieving pixel-level similarity was challenging.

### E.2.2 THE HYPERPARAMETERS

We set the inference steps to 50 with a cfg coefficient of 2.5, following SV3D's configuration, and utilize the Euler sampling method for denoising. Noise optimization comprises 20 steps using a gradient descent optimizer with a learning rate of 1500 and a momentum of 0.5.

### E.3 PERFORMANCE EVALUATION

We mainly use Perceptual Similarity (LPIPS (Zhang et al., 2018)), Structural SIMilarity (SSIM (Wang et al., 2004)), and CLIP similarity score (CLIP-S (Ilharco et al., 2021)) to measure the quality of generated results. Due to the lack of multi-view ground truth, pixel-level evaluation metric, such as PSNR, is not applicable.

The quantitative results in Table 8 demonstrate that optimized noises lead to higher image-to-3D generation quality.

To facilitate a more intuitive comparison, we also present the qualitative results of original noises and optimized noises in Figure 10. It illustrates the significant difference between the optimized noise and the original noise. We can observe that the 3D objects of optimized generally exhibit fewer jagged edges, smoother surfaces, and better fidelity than the results of original noises.

Table 8: The quantitative results of noise optimization for image-to-3D diffusion models according to novel multi-view synthesis on OmniObject3D static orbits.

| Model | Noise | LPIPS↓ | SSIM↑ |
|---|---|---|---|
| SV3D-U | Original Noise | 0.2538 | 0.8664 |
| | Optimized Noise | **0.2523** | **0.8768** |

