# OpenReview forum: "Not All Noises Are Created Equally: Diffusion Noise Selection and Optimization"
_ICLR.cc/2025/Conference — ICLR 2025 Conference Withdrawn Submission_

### Official Review · Reviewer_bLiG · 2024-10-24

**Soundness:** 2
**Presentation:** 3
**Contribution:** 2
**Rating:** 5
**Confidence:** 4

**Summary:**

The paper introduces noise inversion stability as a core concept for improving the performance of diffusion models. It proposes two algorithms, **noise selection** and **noise optimization**, which aim to enhance the quality of generated images by selecting and optimizing initial noise based on a noise-inversion stability metric. The work demonstrates improvements on multiple datasets.

**Strengths:**

1. **Noise Inversion Stability as a Novel Metric**:
   The introduction of **noise inversion stability** as a noise quality metric is well-motivated. The use of **cosine similarity** for measuring stability and the detailed analysis in **Appendix 2** show the compact and thoughtful experimental setup. This approach highlights the significance of noise selection.

2. **Clear Presentation and Motivational Framing**:
   The paper is well-written, with clear motivation and smooth transitions between different sections, making it easy to follow. The concept of improving noise stability is intuitive, making it an interesting starting point for further work in the diffusion noise space.

**Weaknesses:**

1. **Overlaps with Existing Work (FPI)**:
    - Applying **fixed-point iteration  (FPI) with gradient descent** is not a novel noise optimization method, a line of previous works [1, 2, 3] also implemented fixed-point optimization for enhancing noise quality. While the authors claim novelty in their noise optimization approach, it closely resembles other works, with the main difference being the loss function. For example, FPI [1] uses **MSE**, while this work relies on **cosine similarity**. The paper does not adequately discuss these overlaps, nor does it clearly differentiate its contribution from existing work.
   - Additionally, **VISII** [4] has previously used cosine similarity for optimizing condition embeddings, further reducing the originality of this optimization approach. Without clearer distinction, it’s difficult to view this method as a significant leap forward.

2. **Limited Depth in Noise Selection Analysis**:
   The **noise selection** method presented in the paper seems relatively simplistic. It mainly involves selecting from 100 noise seeds based on cosine similarity, which feels like a **naïve approach**. Prior work, such as **D-Flow** [5], provided a more comprehensive analysis of noise selection in controllable image generation. In contrast, this work does not offer an in-depth exploration of  noise selection, such as heuristic exploration in the noise space, or how the number of noise seeds influences the trade-off between efficiency and quality.

3. **Unfair Comparison with DOODL and Lack of Broader Baselines**:
   - I think that the comparison with **DOODL** (**Table 6**) is problematic due to the difference in base models. The paper uses **SDXL** in its experiments, while DOODL was evaluated using **Stable Diffusion 1.4**. This introduces an unfair baseline, as SDXL represents a newer, more powerful model.
   - Furthermore, there is a lack of comparison with other relevant noise selection (*e.g.*, Try, Try Again [6]) and noise optimization techniques (*e.g.*, ReNoise [1], AIDI [2], FPI [3]).

**Reference**

[1] *Garibi, et al. "ReNoise: Real Image Inversion Through Iterative Noising." ECCV (2024).*

[2] *Pan, et al. "Effective real image editing with accelerated iterative diffusion inversion." ICCV (2023).*

[3] *Meiri, et al. "Fixed-point Inversion for Text-to-image diffusion models." arXiv (2023).*

[4] *Thao, et al. "Visual instruction inversion: Image editing via visual prompting." NeurIPS (2023)*

[5] *Ben-Hamu, et al. "D-Flow: Differentiating through Flows for Controlled Generation." ICML (2024).*

[6] *Shyamgopal, et al., Try, Try Again: Faithful diffusion-based text-to-image generation by selection. arXiv (2023).*

**Questions:**

1. **Clarification of Eq. 2**:
   - **Eq. 2** provides an approximation of DDIM inversion process, the exact inversion process should be:
$$
x_t^* = \frac{\sqrt{\alpha_t}}{\sqrt{\alpha_{t-1}}} x^*_{t-1} + \sqrt{\alpha_t} \Big( \sqrt{\frac{1}{\alpha_t} - 1} -  \sqrt{\frac{1}{\alpha_{t-1}} - 1} \Big) \mu_\theta(x_{t-1}^*, t-1, c).
$$
   - Specifically, in my view, the coefficient of $\mu_{\theta}(x_{t-1}, \cdot) $ in **Eq. 2**,  should be $\sqrt{\alpha_{t}}$ rather than $\sqrt{\alpha_{t-1}}$. Could the authors clarify if this replacement is intentional as an approximation? If so, it would be useful to further explain the reasoning and extend the analysis, especially considering the importance of rigor in such approximations.
   - Additionally, it seems the authors apply another approximation to the time condition by replacing $\mu_\theta(x_{t-1}^*, t-1)$ with $\mu_\theta(x^*_{t-1}, t)$. If this approximation is intentional, it would be helpful to explicitly mention it for the sake of clarity.

2. **Comparison with Other Noise Optimization Techniques**:
   - Can you clarify their claims of novelty in light of these existing works (ReNoise [1], AIDI [2], FPI [3])?
   - Can you extend a quantitative comparison with employing strategies in other noise optimization techniques, such as ReNoise [1], AIDI [2], FPI [3]?

3. **Broader Comparison / In-depth Analysis on Noise Selection**:
   - Can you provide comparison with other noise selection methods, such as **Try, Try Again** [6], and extend **DOODL** within the same base model (*e.g*., SDXL) to ensure comparison  fairness.
   - What about conducting ablation studies on the number of noise seeds (K=100 in the manuscript) or to explore more sophisticated selection strategies, and explain how this could potentially improve the selection?

4. **Efficiency Concerns**:
   - I’m concerned about the computational efficiency of the proposed approach. The paper mentions selecting from 100 random noise seeds for noise selection, which adds memory overhead, and the noise optimization process also introduces **time costs**. Can the authors provide a **quantitative comparison** of the **memory** and **time cost** between their method and baseline models (*e.g*., vanilla **SDXL**), such as GPU memory usage, inference time per image, and any additional preprocessing time required? This would help assess the practical feasibility of the proposed methods.

---

### Official Review · Reviewer_s3M3 · 2024-10-28

**Soundness:** 3
**Presentation:** 3
**Contribution:** 2
**Rating:** 5
**Confidence:** 5

**Summary:**

The paper proposes a novel approach for image generation based on noise (seed) selection and optimization. The authors observe that seeds with high inversion accuracy correspond to high-quality outputs, exhibiting improvements in color, style, text rendering, object co-occurrence, positioning, and counting. They also demonstrate an application for 3D generation. The method is evaluated on three popular generation benchmarks.

**Strengths:**

- The concept of evaluating noise seeds based on their inversion quality is both interesting and novel.
- The paper is well-written and well organized.
- There is a comprehensive evaluation across multiple benchmarks, including human assessments.
- The quantitative and qualitative results are convincing.
- The application for 3D generation is intriguing and opens up potential new research directions.

**Weaknesses:**

- The authors present evidence suggesting a strong correlation between inversion stability scores and image quality, where higher stability scores lead to better quality. However, this claim is based on a single example, without theoretical backing or a systematic study. Although experiments indicate that finding fixed points improves quality, it is unclear why this approach is preferable over optimizing a different objective. A theoretical analysis and a comprehensive empirical study—such as testing 100k seeds to show that high s(\epsilon)scores correlate with high image quality—would strengthen this claim.
- The authors chose DDIM inversion as their inversion method; however, studies [1,2,3,4] have shown that DDIM inversion suffers from errors at each time step due to the approximation of the implicit function. These errors accumulate, leading to inconsistencies between the forward and backward processes and ultimately resulting in poor inversion quality. More effective inversion techniques, such as Fixed-point inversion [2,3] or Gradient-descent-based inversion [4], yield improved results.
- Both the noise selection algorithm and noise optimization demand significantly more computational resources and time compared to standard generation. It might not be necessary to generate the entire image; evaluating s(\epsilon) during the first denoising step could be sufficient to determine if the seed is stable.
- Missing citations [5,6].
- The authors claim that stable and optimized seeds produce better image quality than random seeds; however, FID scores are not provided to support this.

[1] Mokady,. et al. (2023). *Null-text Inversion for Editing Real Images using Guided Diffusion Models*.
[2] Pan., et al. (2023). *Effective Real Image Editing with Accelerated Iterative Diffusion Inversion*.
[3] Meiri., et al. (2023). *Fixed-point Inversion for Text-to-Image Diffusion Models*.
[4] Hong., et al. (2023). *On Exact Inversion of DPM-Solvers*.
[5] Samuel., et al. (2024). *Generating Images of Rare Concepts using Pre-trained Diffusion Models*.
[6] Chen., et al. (2024). *TiNO-Edit: Timestep and Noise Optimization for Robust Diffusion-Based Image Editing*.

**Questions:**

See weaknesses. My main concerns are the lack of theoretical or empirical support for your approach and the reliance on DDIM inversion as the inversion method.

---

### Official Review · Reviewer_aBt4 · 2024-10-31

**Soundness:** 3
**Presentation:** 3
**Contribution:** 2
**Rating:** 5
**Confidence:** 3

**Summary:**

The autors hypothesize a correlation between the inversion stability and the quality of diffision models generated images. It uses that matric for two use cases: noise selection and noise optimization

**Strengths:**

- The basic hypothesis of correlation between "inversion stability",  as defined by the authors, and the quality of DM results make a lot of sense
- The suggested methods of noise selection and noise cancelation directly optimized the inversion stability metric and backed up with the results
- The suggested method is not biased by external evaluators

**Weaknesses:**

- The suggested method is sub-optimal by definition since its strongly biased by the inversion process , for example a Noise A may results better quality then Noise B, but the Inversion process of Image A may yield poor inversion stability score due to the inversion process.

**Questions:**

I would suggest to do a comparisons of different inversion methods and their effect on the results

---

### Official Review · Reviewer_14zt · 2024-11-03

**Soundness:** 1
**Presentation:** 2
**Contribution:** 1
**Rating:** 3
**Confidence:** 5

**Summary:**

This paper aims to enhance the quality of images generated by diffusion models by modifying the noise seed. Improved image quality is reflected in better alignment with the input prompt and fewer artifacts. The process involves sampling various noise seeds and either selecting the most suitable one or using optimization techniques. The selection and optimization methods rely on the hypothesis that the noise at each time step during the forward process (image generation) and the backward process (inversion) should be similar, measured using cosine loss.

**Strengths:**

The examples presented in the paper are interesting, suggesting that noise optimization could potentially enhance image quality. The paper also includes results for image-to-3D generations. The proposed method is straightforward and easy to implement.

**Weaknesses:**

I am afraid that I have some fundamental concers about this paper.

1.	I believe that the claim that this is the first paper to study the noise space for image generation is not accurate, as there are at least three prior works that explore this topic [1, 2, 3] ([3] is cited in the paper).
2.	Presentation should be improved. Some references appear unrelated, such as those discussing optimizing noise with additional information ...  (Kirstain et al., 2023; Xu et al., 2024), in line 046. These references pertain to datasets rather than methods. Section 5 should be part of the introduction.
3.	The paper's novelty is limited; studying noise space and optimizing noise seeds has been explored in previous notable works [1-3]. In these works, the diffusion model is also fixed, and optimization directly targets the noise. Considering that noise optimization has already been done in diffusion models (with frozen weights) and inversion techniques, and given the impracticality of sampling and selecting the best noise, the only new contribution appears to be the hypothesis of noise inconsistency and use of cosine loss for noise optimization.
4.	Theoretical issue-1: DDIM is well known to be a poor method for inversion suffering from accumulated errors, as studied in many previous studies  e.g. [4,5,6] . So the inconsistency is particularly linked to DDIM approach and the associated approximation (using $x_{t-1}$ instead of $x_t$, in Eq. 2), are inherent to DDIM, and not to the diffusion model itself. There are more accurate inversion models such as [4,5,6]. Better can also consistency can also come from lower difference between $x_t$ and  $x_{t-1}$, which can be property of the generated image and where the DDIM approximation is better. It would be informative to see the results using better inversion methods like [4-6].
5.	The evaluation is not sufficient: There is no comparison to other methods. In section 5 the authors present alternative approaches and claim that they can’t be compared. I  respectfully disagree. There should be a comparison with the mentioned relevant papers and then discussion over the pros and cons of different methods. I also believe that quantitative measures such as FID and IS are still valuable since they can be done on mass (many images) and should come in addition to the user study. The  improvement indicated in Table 1 seems to be marginal, often around 1%, making it difficult to perceive significant progress.
6.	Claims regarding the counting and positioning accuracy of objects should be validated through experiments rather than a limited number of demonstrations. The paper’s claim about object count and position is not supported by the human evaluation, as Appendix A, page 17, does not address these aspects in its questions.
7.	High computational cost: As noted in Section 5, the proposed method requires generating each image using K forward and backward passes, effectively increasing the computation by factor 2K. With K set to 100, as done in the paper, this results in a 200-fold increase in runtime.

[1] D. Samuel etal, Generating images of rare concepts using pre-trained diffusion models, AAAI 2024

[2] D. Samuel etal, Norm-guided latent space exploration for text-to-image generation, NeurIPS 2023

[3] Xiefan Guo etal, Initno: Boosting text-to-image diffusion models via initial noise optimization. CVPR, 2024.

[4] Mokadt et al, Null-text Inversion for Editing Real Images using Guided Diffusion Models, CVPR 2023

[5] Zhang etal,  Exact Diffusion Inversion via Bi-directional Integration Approximation, CVPR 2024

[6] Pan etal, Effective real image editing with accelerated iterative diffusion inversion, ICCV 2023

**Questions:**

How this cosine is measured? Epsilon  is generally not a vector, it is a tensor?

---

### Note · Authors · 2024-11-13

**Comment:**

Thank you for the thorough review and constructive feedback. We will continue to refine and improve our work based on your suggestions.

Once again, thank you.

**Withdrawal Confirmation:**

I have read and agree with the venue's withdrawal policy on behalf of myself and my co-authors.